

# The bacterial strains JAM1$^T$ and GP59 of the species *Methylophaga nitratireducenticrescens* differ in their expression profiles of denitrification genes in oxic and anoxic cultures

Livie Lestin and Richard Villemur

Centre Armand-Frappier Santé Biotechnologie, Institut National de la Recherche Scientifique, Laval, Québec, Canada

## ABSTRACT

**Background**. Strain JAM1$^T$ and strain GP59 of the methylotrophic, bacterial species *Methylophaga nitratireducenticrescens* were isolated from a microbial community of the biofilm that developed in a fluidized-bed, methanol-fed, marine denitrification system. Despite of their common origin, both strains showed distinct physiological characters towards the dynamics of nitrate ($NO_3^-$) reduction. Strain JAM1$^T$ can reduce $NO_3^-$ to nitrite ($NO_2^-$) but not $NO_2^-$ to nitric oxide (NO) as it lacks a NO-forming $NO_2^-$ reductase. Strain GP59 on the other hand can carry the complete reduction of $NO_3^-$ to $N_2$. Strain GP59 cultured under anoxic conditions shows a 24-48h lag phase before $NO_3^-$ reduction occurs. In strain JAM1$^T$ cultures, $NO_3^-$ reduction begins immediately with accumulation of $NO_2^-$. Furthermore, $NO_3^-$ is reduced under oxic conditions in strain JAM1$^T$ cultures, which does not appear in strain GP59 cultures. These distinct characters suggest differences in the regulation pathways impacting the expression of denitrification genes, and ultimately growth.

**Methods**. Both strains were cultured under oxic conditions either with or without $NO_3^-$, or under anoxic conditions with $NO_3^-$. Transcript levels of selected denitrification genes (*nar1* and *nar2* encoding $NO_3^-$ reductases, *nirK* encoding $NO_2^-$ reductase, *narK12f* encoding $NO_3^-/NO_2^-$ transporter) and regulatory genes (*narXL* and *fnr*) were determined by quantitative reverse transcription polymerase chain reaction. We also derived the transcriptomes of these cultures and determined their relative gene expression profiles.

**Results**. The transcript levels of *nar1* were very low in strain GP59 cultured under oxic conditions without $NO_3^-$. These levels were 37 times higher in strain JAM1$^T$ cultured under the same conditions, suggesting that Nar1 was expressed at sufficient levels in strain JAM1$^T$ before the inoculation of the oxic and anoxic cultures to carry $NO_3^-$ reduction with no lag phase. Transcriptomic analysis revealed that each strain had distinct relative gene expression profiles, and oxygen had high impact on these profiles. Among denitrification genes and regulatory genes, the *nnrS3* gene encoding factor involved in NO-response function had its relative gene transcript levels 5 to 10 times higher in strain GP59 cultured under oxic conditions with $NO_3^-$ than those in both strains cultured under oxic conditions without $NO_3^-$. Since NnrS senses NO, these results suggest that strain GP59 reduced $NO_3^-$ to NO under oxic conditions, but

Corresponding author
Richard Villemur,
richard.villemur@inrs.ca

because of the oxic environment, NO is oxidized back to $NO_3^-$ by flavohemoproteins (NO dioxygenase; Hmp), explaining why $NO_3^-$ reduction is not observed in strain GP59 cultured under oxic conditions.

**Conclusions**. Understanding how these two strains manage the regulation of the denitrification pathway provided some clues on how they response to environmental changes in the original biofilm community, and, by extension, how this community adapts in providing efficient denitrifying activities.

# INTRODUCTION

Members of the genus *Methylophaga* are *gammaproteobacteria* that are common in marine or brackish water. As methylotrophic bacteria, they only use one-carbon compounds, for instance methanol, methylamine or dimethyl sulfide, as carbon and energy sources for methylotrophic growth (*Boden, 2012*; *Boden, 2019*). From a naturally occurring biofilm that developed in a fluidized-bed, methanol-fed denitrification system operating in continuous mode, and treating seawater tank at the natural science museum of the Montreal Biodome (*Labbé et al., 2003*; *Labbé, Parent & Villemur, 2003*; *Parent & Morin, 2000*; *Labbé et al., 2007*; *Laurin et al., 2008*), two strains belonging to the species *Methylophaga nitratireducenticrescens* were isolated: strain JAM1[T] and strain GP59 (*Auclair et al., 2010*; *Villeneuve et al., 2013*; *Mauffrey, Martineau & Villemur, 2015*; *Mauffrey et al., 2017*; *Geoffroy et al., 2018*). These strains are the only known *Methylophaga* species capable to grow under anoxic conditions with nitrate ($NO_3^-$) as terminal electron acceptor. Strain GP59 possesses the complete denitrification pathway, whereas strain JAM1[T] lacks a NO-forming nitrite ($NO_2^-$) reductase activity. Their genomes have about 90% identity in nucleic acid sequences, and share the same denitrification island, a 66.5 kb region containing operons or gene clusters encoding two Nar-type $NO_3^-$ reductases (Nar1 and Nar2 systems; EC 1.7.5.1), two nitric oxide (NO) reductases (Nor1 and Nor2 systems; EC 1.7.2.5) and one nitrous oxide ($N_2O$) reductase (Nos system; EC 1.7.2.4). This region also encodes three $NO_3^-$ transporters (NarK1, NarK2, NarK12f) and regulatory factors such as NarX/NarL, NosR and NorRE, and NnrS involved in NO-response. Whereas no gene encoding a NO-forming $NO_2^-$ reductase (NirK- or NirS-type) is present in strain JAM1[T] genome, a gene encoding the $NO_2^-$ reductase NirK (EC 1.7.2.1) is found in another region of strain GP59 genome. Based on the repeat pattern found in their respective CRISPR locus, one strain did not evolve recently from the other but rather originate from a common ancestor (*Geoffroy et al., 2018*).

We showed in previous reports (*Geoffroy et al., 2018*; *Payette et al., 2019*; *Villemur et al., 2019*) that strain GP59 and strain JAM1[T] have different growth dynamics in the biofilm community upon environmental changes. In the original biofilm in the Biodome denitrification system, the proportion of strain JAM1[T] was higher than that of strain GP59.

However, when this biofilm was cultured under laboratory-scale anoxic conditions (in vials, batch-fed mode instead of continuous mode, with artificial seawater medium instead of the commercial Instant Ocean medium), strain GP59 increased dramatically in proportion in the biofilm, while the level of strain JAM1$^T$ stayed the same. From these accumulated data, we believe that those two subpopulations of *M. nitratireducenticrescens* display distinct physiological characters in response to environmental changes in the biofilm.

Pure cultures of strain JAM1$^T$ and strain GP59 also demonstrate different dynamics in their denitrification activities that impact their growth. Under anoxic conditions, strain GP59 requires a lag time of 24 to 48 h before $NO_3^-$ reduction occurs. Such lag time is not apparent in strain JAM1$^T$ anoxic cultures and $NO_3^-$ begins to be reduced almost immediately (*Geoffroy et al., 2018*). Besides, strain GP59 cultures generate higher biomass yield than strain JAM1$^T$ under anoxic conditions because of the completeness of $NO_3^-$ reduction to $N_2$, unlike strain JAM1$^T$ where toxic $NO_2^-$ accumulates in the medium. Both strains respond differently when cultured under oxic conditions in presence of $NO_3^-$. Although cultures from both strains generate equivalent growth yield under oxic conditions, strain JAM1$^T$ cultures can reduce $NO_3^-$ to $NO_2^-$ (*Mauffrey et al., 2017*), which does not appear in strain GP59 cultures (*Geoffroy et al., 2018*). Among possible mechanisms explaining these distinct behaviors, we hypothesized that one of them involves the control of the gene expression of the denitrification pathway that operates differently in both strains.

The control of denitrification is carried out by a complex network at the transcriptional level. This network involves CRP/FNR family transcriptional regulators and specific two-component systems, in response to various signals perceptible in the bacterial environment (*Körner, Sofia & Zumft, 2003*; *Gaimster et al., 2017*; *Durand & Guillier, 2021*). For instance, FNR intervenes in response to low level of oxygen, in *E. coli*, by controlling hundreds of genes, including the *nar* operon (*narGHJI*), by binding in upstream sequence (*fnr* box) of target genes (*Zumft, 1997*; *Körner, Sofia & Zumft, 2003*; *Constantinidou et al., 2006*; *Van Spanning, Richardson & Ferguson, 2007*). NarX/NarL is a two-component system that senses the presence of $NO_3^-$ by stimulating the expression of the *nar* operon. Putative FNR and NarL nucleic acid binding sites upstream of *narXL* and of both *nar* operons were found in strain JAM1$^T$ genome (*Mauffrey, Martineau & Villemur, 2015*).

Besides genes encoding NorRE and NosR that regulate the expression of genes encoding the Nor and Nos systems, respectively, open reading frames (ORFs) encoding putative regulators or proteins with NO-response function were found in both genomes, which include NnrS involved in response to NO, the $NO_2^-$-sensitive transcriptional repressor NsrR (Rrf2 family transcriptional regulator), flavohemoprotein (NO dioxygenase; *hmp*) (EC 1.14.12.17), and DnrN/YtfE known to be involved in iron-sulfur cluster repair di-iron protein or to be NO-dependent regulator (*Körner, Sofia & Zumft, 2003*; *Spiro, 2011*; *Stern et al., 2013*; *Guo & Gao, 2021*).

To test our hypothesis, we aimed by different culture conditions to measure the expression levels of key genes that could have impacted the denitrifying activities of these strains. Pure cultures from both strains were performed to get synchronized culture replicates under oxic or anoxic conditions with $NO_3^-$. The expression levels of selected

genes were measured early in culture growth with minimal $NO_3^-$ reduction to assess early regulation of denitrification genes, or halfway through of this reduction to assess the evolution of this regulation. Total RNA was extracted from culture replicates, and by using reverse transcription quantitative polymerase chain reaction (RT-qPCR) and RNA sequencing, we compared the expression levels of genes involved in the denitrification pathway between these types of cultures for both strains. Deciphering the regulation mechanisms of both strains will shed some light on how they response to environmental changes in the original biofilm community. By extension, our results will provide new knowledge on the dynamism of species' subpopulations in other bioprocesses.

## MATERIAL AND METHODS

### Culture medium and conditions

*M. nitratireducenticrescens* strains JAM1[T] and GP59 were cultured in the *Methylophaga* medium 1403 (per 970 mL: 24 g NaCl, 3 g $MgCl_2.6$ $H_2O$, 2 g $MgSO_4.7H_2O$, 0.5 g KCl, 1 g $CaCl_2$, 0.5 g Bistris, pH 8.0). This medium was autoclaved before the addition of sterilized solutions: methanol (three mL per liter medium), stock solution T (20 mL per liter medium), 0.1 mg $mL^{-1}$ vitamin $B_{12}$ (one mL per liter medium), and stock Wolf's mineral solution (10 mL per liter medium). When needed, the media was supplemented with 21.4 mM sodium $NO_3^-$ (final concentration). Stock solution T was made of (per 100 mL): 0.7 g $KH_2PO_4$, 10 g $NH_4Cl$, 10 g Bistris, 0.3 g Ferric ammonium citrate, pH 8. Stock Wolf's mineral solution was made of (per 1000 mL): 0.5 g EDTA, 3 g $MgSO_4.7H_2O$, 0.5 g $MnSO_4.H_2O$, 1 g NaCl, 0.1 g $FeSO_4.7H_2O$, 0.1 g $CoCl_2.6H_2O$, 0.1 g $CaCl_2$, 0.1 g $ZnSO_4.7H_2O$, 0.01 g $CuSO_4.5H_2O$, 0.01 g $AlK(SO_4)_2.12H_2O$, 0.01 g $H_3BO_3$, 0.01 g $Na_2MoO_4.2H_2O$ [from American Type Culture Collection, Manassas VI, USA]. The final concentration of $NH_4^+$ was 3.8 mM.

Cultures were carried out under three conditions: anoxic with 21.4 mM $NO_3^-$ (here named <<AN>> conditions), oxic with 21.4 mM $NO_3^-$ (here named <<ON>> conditions) and oxic without $NO_3^-$ (here named <<O>> conditions). These conditions were chosen to assess the impact of oxygen (presence or absence) on the gene expression patterns in both strains. We set two different sampling times in the <<AN>> cultures: one with minimal $NO_3^-$ reduction (<10%) here named "*Low period*" and the other one during high rate of $NO_3^-$ reduction (20 to 50% $NO_3^-$ reduction) here named "*High period*".

Anoxic cultures (<<AN>> conditions) were carried out in 30-mL (*High period*) or 300-mL (*Low period*) medium in 70-mL or 500-mL serum vials, respectively. Bottles were sealed with rubber stoppers maintained by a metal ring or sterile septum caps, and medium was purged for 10-15 min with pure nitrogen gas, prior to sterilization. Oxic cultures (<<O>> and <<ON>> conditions) were carried out in 30-mL (*High period*) or 300-mL (*Low period*) medium in 250-mL or 1000-mL Erlenmeyer flasks, respectively.

The bacterial inoculum was made of a fresh oxic culture without $NO_3^-$ (<<O>> conditions). All cultures were inoculated at a final optic density 600 nm ($OD_{600}$) between 0.05 and 0.1, and incubated at 30 °C, under a constant agitation at 150 rpm. Samples were taken to monitor growth by spectrophotometry ($OD_{600}$). These samples were homogenized

**Table 1  Concentration of $NO_3^-$ that was reduced during the *Low* and *High periods*.**

| Culture conditions | Phase | $NO_3^-$ (mM) | | | | Growth ($OD_{600}$) | | | |
| | | JAM1[T] | | GP59 | | Final | | | |
| | | Initial | Final | Initial | Final | JAM1[T] | Time | GP59 | Time |
|---|---|---|---|---|---|---|---|---|---|
| <<AN >> | Low | 15.7 (0.2) | 14.7 (0.2) | 22.7 (0.5) | 21.3 (0.2) | 0.08 | 0.5 h | 0.09 | 8 h |
| <<ON >> | Low | 21.0 (0.1) | 19.8 (0.5) | 18.8 (0.0) | 18.7 (0.0) | 0.08 | 0.5 h | 0.08 | 1 h |
| <<O >> | Low | NA | NA | NA | NA | 0.07 | 0.5 h | 0.09 | 1 h |
| <<AN >> | High | 16.2 (0.0) | 12.3 (0.0) | 17.7 (0.1) | 12.9 (0.3) | 0.27 | 15 h | 0.34 | 96 h |
| <<ON >> | High | 12.3 (0.0) | 8.9 (0.2) | 16.8 (0.1) | 17.2 (0.1) | 0.88 | 13 h | 0.53 | 14 h |
| <<O >> | High | NA | NA | NA | NA | 0.91 | 15 h | 0.51 | 15 h |

Notes.

Cultures were inoculated at 0.05 to 0.1 $OD_{600}$, and samples were immediately taken to measure $NO_3^-$ (Initial). After prescribed times (estimated from prior culture tests), the whole cultures were taken (Final) to collect the biomass and to measure $NO_3^-$ and $OD_{600}$. Values are from triplicate cultures with standard deviation under parentheses.

<<AN >>, anoxic conditions; <<O >>, oxic conditions, no $NO_3^-$; <<ON >>, oxic conditions with $NO_3^-$; NA, not applicable.

using a potter-Elvehjem homogenizer prior to measurements to disperse the flocs formed during the growth. Preliminary cultures were performed to determine the correspondence between the culture growth ($OD_{600}$) and the level of remaining $NO_3^-$ in cultures. The dynamics of growth and of $NO_3^-$ and $NO_2^-$ reductions occurred as reported before (*Geoffroy et al., 2018*) (lag phase under the <<AN >> conditions and no $NO_3^-$ reduction under the <<ON >> conditions for GP59 cultures; see Data S1).

We then performed independent cultures (from different inocula and different days), from which total biomass (sacrificed cultures) was collected at prescribed times based on preliminary culture assays, and preserved at −70 °C. The residual concentrations of $NO_3^-$ and $NO_2^-$ in the <<ON >> and <<AN >> cultures were measured later from the supernatants by ion chromatography with the 850 Professional IC (Metrohm, Herisau, Switzerland) or by colorimetric assays described by *Cucaita, Piochon & Villemur (2021)*. From these latter measurements, we chose at least three frozen samples that fit our criteria of $NO_3^-$ reduction to perform RNA extraction. The chosen cultures of strain JAM1[T] cultured under the <<AN >> and <<ON >> conditions, and cultures of strain GP59 cultured under the <<AN >> conditions showed around 6% $NO_3^-$ reduction during the *Low period* or around 25% $NO_3^-$ reduction during the *High period* (Table 1). As $NO_3^-$ reduction does not occur in strain GP59 cultured under the <<ON >> conditions, biomass was collected when cultures reached about the same times than strain JAM1[T] cultures at the *Low* and *High periods*. Under the <<O >> conditions, samples for both periods were collected for each strain around the same time as the <<ON >> conditions (Table 1).

## RNA extraction and RT-qPCR assay

The biomass of the 300-mL cultures (*Low period*) was collected by filtration on a 0.22 μm filter. The biomass of the 30-mL cultures (*High period*) was collected by centrifugation at $8000 \times g$ for 10 min at 4 °C. The pellets or filters were transferred in 2-mL tubes containing 250 mg of 0.2 mm glass beads, then one mL of extraction buffer (50 mM Tris–HCl, 100 mM EDTA, 150 mM NaCl pH 8.0) and one mL of water-saturated phenol (pH 4.3) were added.

**Table 2  Primers used for RT-qPCR assays.**

| Name | Sequence (5′–3′) | Hybridization Temp (°C) | Length nt | References |
|---|---|---|---|---|
| Standard genes | | | | |
| rpoD (10F) | CAGCAATCACGCGTTAAAGA | 60 | 144 | *Mauffrey, Martineau & Villemur (2015)* |
| rpoD (153R) | ACCCAGGTCGCTGAACATAC | | | |
| rpob (3861F) | TGAGATGGAGGTTTGGGCAC | 60 | 146 | *Mauffrey, Martineau & Villemur (2015)* |
| rpob (4006R) | GCATACCTGCATCCATCCGA | | | |
| dnaG (774F) | CATCCTGATCGTGGAAGGTT | 60 | 121 | *Mauffrey, Martineau & Villemur (2015)* |
| dnaG (894R) | GCTGCGAATCAACTGACGTA | | | |
| Regulatory genes | | | | |
| narX1 (1403F) | TGCTGAAGCCCTACAAGTGG | 60 | 133 | This study |
| narX1 (1535R) | TGCGTTAGCGATAGCACCTT | | | |
| narL1 (174F) | ATGCCGGGAATAGGAGGAGT | 60 | 136 | This study |
| narL1 (309R) | AATAACCGCGGGCACCATTA | | | |
| fnr2 (121F) | ACCGGCTATGTCTACCGTTG | 60 | 149 | This study |
| fnr2 (269R) | CGAGCCTGAGCGAACAACAA | | | |
| Selected denitrification genes | | | | |
| qnarG1-F | AGCCCACATCGTATCAAGCA | 61 | 149 | *Geoffroy et al. (2018)* |
| qnarG1-R | CCACGCACCGCAGTATATTG | | | |
| narK12 (257F) | TTCTGATCTGCCCGAACTCT | 60 | 106 | *Mauffrey, Martineau & Villemur (2015)* |
| narK12 (362R) | GCGCCTAGCAATGCTTTTAC | | | |
| narG2 (597F) | TTACGCTGCAGGATCACGTT | 60 | 127 | *Mauffrey, Martineau & Villemur (2015)* |
| narG2 (723R) | TGACTCGGGTACATCGGTCT | | | |
| qnirK-F | AAGTCGGTAAAGTAGCCGTTGA | 55 | 138 | *Geoffroy et al. (2018)* |
| qnirK-R | TCTCCATCGTCATTTGAACAAC | | | |

The samples were flash frozen in liquid nitrogen and stored at −70 °C until extraction. Sample collection of the anoxic cultures was carried out in controlled airtight chamber flushed with nitrogen gas.

The RNA extraction was previously described by *Mauffrey, Martineau & Villemur (2015)*. Because low amount of biomass was produced by the *Low period* cultures (despite the 300-mL cultures), total RNA extracted from these cultures allowed us to carry all RT-qPCR assays, but not the transcriptomes. The *High period* cultures generated much higher amount of biomass, and thus RNA, allowing to carry RT-qPCR assays and transcriptomes from replicate cultures. RNA quality was verified by agarose gel electrophoresis (Fig. S1) and by spectrophotometry (Nanodrop) with ratio $OD_{260}/OD_{280}$ >1.8. The absence of residual DNA in the RNA samples was verified by end-point PCR using the same primers chosen for the RT-qPCR assays (Table 2). cDNA was synthetized with the Reverse Transcription System according to the manufacturer (Promega, Madison, WI, USA), using 800 ng of RNA and hexameric primers. A reaction with no template was made as negative control. cDNA samples were preserved at −70 °C.

Quantification of amplified PCR products was performed with the PerfeCTa® SYBR® Green SuperMix ROXTM (Quanta BioSciences) in the C1000 Touch Thermal cycler real-time PCR machine (Bio-Rad Laboratories, Hercules, CA, USA). The mix for one reaction was composed of 10 µL PerfeCTa® SYBR® Green SuperMix, 40 pmoles of each primer (Table 2), 25 ng cDNA and RNA-free water in a final volume of 20 µL. Primer sequences (Table 2) for *narX* (Q7A_444), *narL* (Q7A_445), *fnr* (Q7A_307), *narG1* (Q7A_446), *narG2* (Q7A_484), *narK12f* (Q7A_479) and *nirK* (CDW43_15165, strain GP59) genes were retrieved from the strain JAM1[T] genome (GenBank accession number CP003390.3) and strain GP59 genome (CP021973.1). Except for *nirK*, the amplified region of respective genes has identical sequences in both genomes. Three reference genes (*Rocha, Santos & Pacheco, 2015*) were selected and used for the data normalization: *rpoD* (sigma factor, GenBank accession number Q7A_343), *rpoB* (RNA polymerase $\beta$ subunit, Q7A_2329), *dnaG* (primase, Q7A_342). The program run 5 min at 95 °C, then 40 cycles: hybridization temperature (Table 2) for 15 s, 20 s at 72 °C and 10 s at 95 °C. The specificity of the reaction was verified by collecting the fluorescence through a melting curve by raising the temperature between 65 and 95 °C. The respective genome was used as template to derive the standard curve. Standard curves were performed with 10-fold serial dilutions in PCR grade water, resulting in a concentration gradient of $10^7$ to $10^0$ copies per reaction. The experiment was validated with efficiency values >85% and $r^2$ values between 0.90 and 1.1. No template controls were performed, with results ranging from 0 to 2 gene copies per reaction. Inhibition was verified by amplification of the normalization genes.

For both strains in each period and culture conditions, RNA was extracted from at least three cultures. RT-qPCR assays were performed in duplicate for each sample. The number of copies per reaction was normalized with the levels of each normalization gene. The transcript levels were expressed as the number of copies per 100 copies of normalized genes. The results of *narX* and *narL* were combined and averaged.

## Sequences analysis

Sequences of the chromosomic region that corresponds to the denitrification island from both strains were aligned (pairwise local alignment) by Bioedit (7.2.3). Searches for DNA binding sites of (i) potential NarL, (ii) factor for inversion stimulation (FIS), (iii) integration host factor (IHF) and (iv) fumarate and $NO_3^-$ reductase regulon (FNR) were carried out with the Virtual Footprint software of PRODORIC (https://bio.tools/prodoric) (*Dudek & Jahn, 2022*).

## Transcriptome analysis

RNA samples retrieved from the three independent cultures at the *High period* were sent for sequencing using the Illumina Method (NovaSeq 6000 S4 PE100). Library preparation and sequencing were performed by the Centre d'expertise et de services Génome Québec (Montréal, QC, Canada). Ribosomal RNA (rRNA) were depleted using the Ribo-ZeroTM rRNA Removal Kit (Meta-Bacteria; Epicentre, Madison, WI, USA). The number of reads per replicate ranged from 6.9 to 78.7 millions. RNAseq reads from strain JAM1[T] and strain GP59 cultured under the <<AN >> conditions were deposited in Sequence Read Archive (SRA;

National Center for Biotechnology Information [NCBI]: https://www.ncbi.nlm.nih.gov/) under the bioproject number PRJNA525230 (SRX5461036, SRX5461037, SRX5461044, SRX5461045, SRX5461047, SRX5461048). For those cultured under the <<O >> and <<ON >> conditions, RNAseq reads were deposited in SRA under the bioproject number PRJNA1072961 (SAMN39755713 to SAMN39755725).

Raw sequencing reads were trimmed using fastq_quality_filter (1.0.2) to remove low quality reads (score < 20) (*Gordon & Hannon, 2010*). The paired reads were then merged and aligned with the reference genome of *M. nitratireducenticrescens* strain JAM1$^T$ or GP59 using Bowtie2 (v 2.5.0) (*Langmead & Salzberg, 2012*), and annotated with Bedtools (v 2.30.0) (*Quinlan & Hall, 2010*). The power analysis calculation (alpha = 0.05, effect = 2) were carried out on all the genes of the triplicate cultures of each condition, using the transcript per million (TPM) values as sequencing depth (online at https://rodrigo-arcoverde.shinyapps.io/rnaseq_power_calc/). The power ranged from 0.96 to 1. Genes that were significantly differentially expressed were identified by EdgeR (v 3.36.0) (*Robinson, McCarthy & Smyth, 2010*) with trimmed mean of M values (TMM) method to normalize library sizes (robust = TRUE; *P*-value adjusted threshold = 0.05; *P*-value adjusted method = Benjamini and Hochberg). All these analyses were performed on the Galaxy server (https://usegalaxy.org/). Genes were considered differentially expressed when the false discovery rate (FDR) was ≤ 0.05.

High proportion of reads that did not align with the strain GP59 genome were found in the RNA samples. RNA preparation from strain GP59 cultures were sequenced at the same time than those of strain JAM1$^T$ cultures that generated <1% unaligned reads to strain JAM1$^T$ genome. In addition, RNA from replicate #3 of strain GP59 cultured under the <<O >> conditions was resequenced and showed again high proportion of unaligned reads. All these results rule out sequence contamination. These "unaligned" reads were assembled *de novo* by Trinity (v. 2.15.1) or Megahit (1.2.9) at the Galaxy server. Estimation of the transcript abundance of the *de novo* assembled sequences was performed (Galaxy server) using RSEM as abundance estimation method (*Li & Dewey, 2011*). Three long transcripts with the highest transcript abundance were examined for putative ORFs with ORF finder (NCBI). ORFs were compared to databases by Blastp at NCBI.

## Statistical analysis

Two-way ANOVA were performed on RT-qPCR with $\log_{10}$-transformed transcript levels with Tukey posttest (GraphPad Prism version 10.2.1). Outliners were identified and removed from the analysis (GraphPad Prism). Relative expression profiles of genes common to strain JAM1$^T$ and strain GP59 were analyzed by Principal coordinate analysis (PCoA) (Canoco version 5.15; *Ter Braak & Šmilauer, 2018*), with Log (1 * X + 1) calculating matrix of distances, using percentage difference (Bray-Curtis distance), and PERMANOVA were performed for significance with 999 permutations (*Anderson, 2017*).

## RESULTS AND DISCUSSION

### Transcript levels of denitrification and regulatory genes

The transcript levels of key genes involved in the denitrification metabolism were measured by RT-qPCR: *narG1* and *narG2* (*nar1* and *nar2* operons; $NO_3^-$ reductases), *narK12f* ($NO_3^-$/$NO_2^-$ transporter), *nirK* ($NO_2^-$ reductase; only strain GP59), *narXL* and a gene encoding a CRP/FNR family transcriptional regulator here named *fnr* (43% identity/66% similarity with *E. coli* FNR in deduced amino acid sequence). The *nar1* polycistronic operon includes genes encoding Nar(1)GHJI, and the NarK1 and NarK2 $NO_3^-$ transporters, as demonstrated by *Mauffrey, Martineau & Villemur (2015)*. The transcript levels of *nor1*, *nor2* and *nos* were not determined because strain JAM1[T] cannot carry out the reduction of $NO_2^-$ to NO, making comparisons between strains of transcript levels for those genes inconclusive.

During the *Low period* (minimum $NO_3^-$ reduction), *narG1* had a very low transcript levels at 1.2 copies per 100 normalized genes (cp/100) in strain GP59 cultured under the $<<O>>$ conditions, compared to 45 cp/100 for strain JAM1[T] ($p < 0.0001$; Fig. 1). The anoxic conditions ($<<AN>>$) stimulated the *narG1* expression in both strains where they reached similar levels (1090 *vs* 1850 cp/100; $p = 0.082$). Such gene expression stimulation was apparent at a lesser extent (62 cp/100) in the $<<ON>>$ cultures for strain GP59, which reached similar levels than those in strain JAM1[T] $<<ON>>$ cultures (80 cp/100). Contrary to GP59 cultures, the *narG1* transcript levels were not different between the $<<O>>$ and $<<ON>>$ conditions (45 *vs* 80 cp/100; $p = 0.128$) in strain JAM1[T] cultures.

*narG2*, *narK12f*, *narXL* and *fnr* had similar transcript levels (3.3 to 17 cp/100) in strain JAM1[T] cultured under the three conditions (Fig. 1). Contrary to strain JAM1[T] cultures, these five genes had higher transcript levels (63 to 123 cp/100) in strain GP59 cultured under the $<<AN>>$ conditions than under the $<<O>>$ and $<<ON>>$ conditions (3.6 to 18 cp/100; Fig. 1). For *nirK* in strain GP59 cultures, its transcript levels were 6 times higher under the $<<AN>>$ conditions (52 cp/100) than under the other conditions (average 8.6 cp/100) (Fig. 1).

During the *High period* (25% $NO_3^-$ reduction), *narG1, narG2, narK12f* and *narXL* had higher transcript levels (2.5 to 8 times) under the $<<AN>>$ conditions compared to the other conditions in strain JAM1[T] cultures (Fig. 1). For the *fnr* transcript levels, strain JAM1[T] showed no significant differences between the three conditions. Surprisingly, the transcript levels of *nirK* were 30 and 5 times higher in strain GP59 cultured under the $<<O>>$ and $<<ON>>$ conditions, respectively, than under the $<<AN>>$ conditions. For the other genes, their transcript levels were at similar levels in strain GP59 cultured under the three conditions (Fig. 1).

The regulation of the expression of the Nar1 system appears to be one of the key elements of the different dynamics of $NO_3^-$ reduction between strain JAM1[T] and strain GP59 during the *Low period*. The $<<O>>$ conditions affected more the expression of the *nar1* operon in strain GP59 cultures, in which its transcript levels were 37.5 times lower than those in strain JAM1[T] cultures. As the inocula of both strains were cultured under the $<<O>>$ conditions, the Nar1 system, including the two $NO_3^-$ transporters (NarK1 and NarK2),

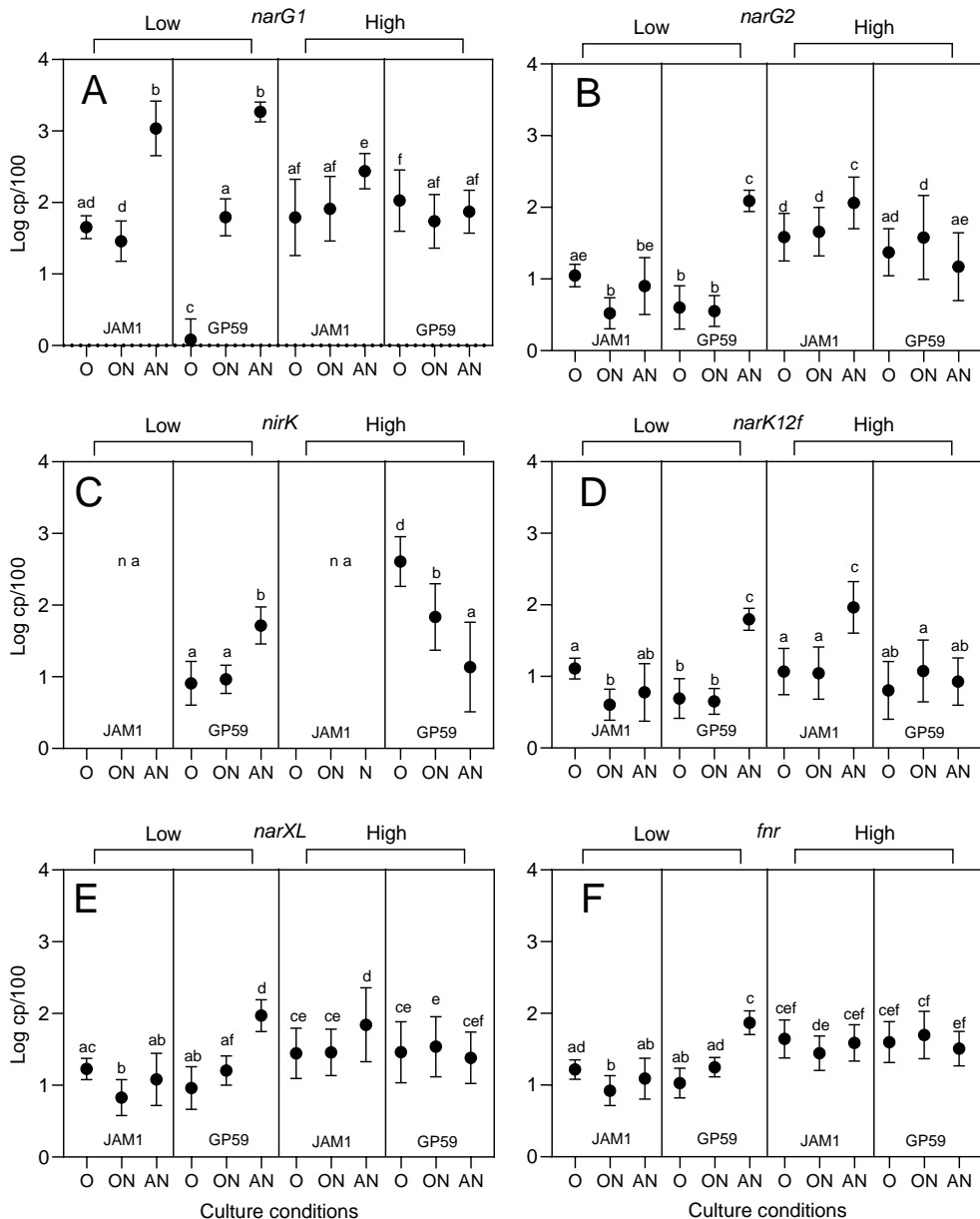

**Figure 1** **Transcript levels of selected denitrification genes and regulatory genes in cultures of strain JAM1[T] and strain GP59.** Strains JAM1[T] and strain GP59 were cultured under ＜＜O＞＞ oxic (＜＜O＞＞), oxic with $NO_3^-$ (＜＜ON＞＞) and anoxic ($NO_3^-$, no oxygen; ＜＜AN＞＞) conditions. Total RNA was extracted from replicate cultures and RT-qPCR were performed. The transcript levels are expressed as the number of gene copies per 100 copies of normalized genes (cp/100). Two-way ANOVA tests with Tukey post hoc tests were performed on $\log_{10}$-transformed transcript levels. Values represented by different letters are highly significantly different ($p < 0.0001$) across culture conditions and strains. Values represented by the same letter are considered not highly significantly different. For instance, in panel A, *Low period*: the ＜＜AN ＞＞ conditions between strain JAM1[T] and strain GP59 showed no difference (same letter "b"). 

**Figure 1 (…continued)**
Strain JAM1$^T$ under the <<O >> conditions (letters "a and d") showed no difference with strain JAM1$^T$ under the <<ON >> conditions (letter "d") and with strain GP59 under the <<ON >> conditions (letter "a"). However, strain JAM1$^T$ under the <<ON >> conditions (letter "d") was different then strain GP59 under the <<ON >> conditions (letter "a"). Data represent mean log$_{10}$ values ± SD. na, not applicable.

was probably already expressed in the inocula of strain JAM1$^T$, which explains no latency in NO$_3^-$ reduction under the <<AN >> and <<ON >> conditions. Because of the very low expression levels of the Nar1 and Nar2 systems under the <<O >> conditions in the strain GP59 inocula, both *nar* operons (or at least one) have to be induced and the reductase(s) and transporters produced under the <<ON >> and <<AN >> conditions for NO$_3^-$ reduction to occur. However, this does not explain why NO$_3^-$ reduction was not observed in strain GP59 cultured under the <<ON >> conditions as it does in strain JAM1$^T$ cultures, because their respective cultures reached the same *nar1* transcript levels under these conditions during the *Low period*.

We did not observe increases of the *fnr* transcript levels under the <<AN >> conditions for the strain JAM1$^T$ cultures compared to the <<O >> and <<ON >> conditions either during the *Low* or the *High period*. This lack of stimulation during the *Low period* correlates with the absence of stimulation of the expression of *narXL*, *narG2* and *narK12f*, but not with the *narG1* expression pattern, where strong stimulation of its expression occurred under the <<AN >> conditions. During the *High period*, although the *fnr* transcript levels did not raise in strain JAM1$^T$ cultured under the <<AN >> conditions, *narXL* did, which in turn stimulated the expression of *narG1*, *narG2* and *narK12f*.

The opposite behavior occurred in strain GP59 cultures where increases of the *fnr* transcript levels under the <<AN >> conditions during the *Low period* concur with increases of those of *narXL*, and the stimulation of the expression of the other genes under these conditions. In addition, during the *High period*, *fnr* did not increase its transcript levels under the <<AN >> conditions, which again concurs with the lack of stimulation of *narG1*, *narG2*, *narXL* and *narK12f* (*nirK* will be discussed later). Our results suggest that the regulation of denitrification genes in strain GP59 follows the expected control by FNR and NarXL observed in other bacterial species.

The level of the Fnr protein in cells does not have to change when the cultures switch from oxic to anoxic conditions as FNR shifts from an inactive (monomer) to active (dimer) configuration (*Mettert & Kiley, 2018*). Deduced amino acid sequences of *fnr* from both strains showed only one amino acid substitution (leucine for isoleucine), and no nucleotide substitution upstream of the *fnr* start codon, which suggests that differences in the expression pattern of the denitrification genes between both strains do not lie on the function of FNR.

## Differences in genome sequences

We compared the nucleic sequences of the denitrification island (66 591 nt; Fig. 2A) between each strain looking for substitutions that may provide some indications explaining the differences in gene expression profiles. We found only 44 nt substitutions (Data S2). None

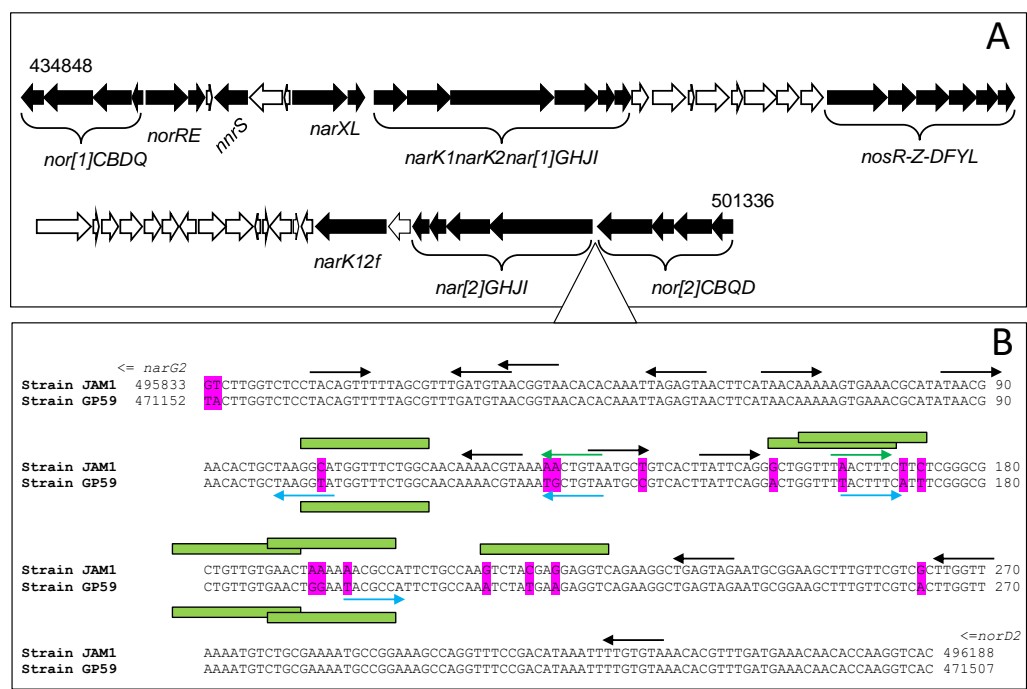

**Figure 2 Chromosomic arrangement of the denitrification island.** (A) Black arrows, Denitrification and regulatory genes. (B) Intergenic sequence (354 nt) between the *nar2* and *nor2* operons. Differences between strain JAM1[T] and strain GP59 sequences are highlighted in magenta. Arrows pointing right represent putative NarL binding sites in the forward sequence; arrows pointing left are sites in the reverse sequence. Blue arrows are putative NarL binding sites specific to strain GP59; green arrows specific to strain JAM1[T]. Black arrows are NarL binding sites common to both strains. Green boxes are putative FIS binding sites (only illustrated in the 100–230 region). Coordinates are from GenBank accession number CP003390.3 for strain JAM1[T] genome and CP021973.1 for strain GP59 genome.

of these substitutions were located upstream of *narXL,* which contains two putative FNR DNA binding sites, and none in the intergenic region between *narXL* and the *nar1* operon. Like *fnr*, differences in the gene expression pattern between both strains do not lie either on the function of NarX/NarL.

The most noticeable changes were found in the intergenic region (354 nt) between the *nor2* and *nar2* operons, where the promotor of the *nar2* operon might be located, with 17 nt substitutions (Fig. 2B). No putative FNR binding site was found, but several potential NarL binding sites were. Among these NarL sites, five of them were affected by one or two substitutions. In addition, several of these substitutions are located in putative FIS (Fig. 2B) and IHF (not illustrated) binding sites. These substitutions may have influenced the interaction between the binding site and NarL (*Maris et al., 2002*; *Maris et al., 2005*). Therefore, higher transcript levels of *narXL* and *fnr* in strain GP59 cultured under the <<AN >> conditions combined with proper NarL binding sites may explain the different *nar2* expression profiles between both strains during the *Low period*.

We showed in a previous work by *Mauffrey, Martineau & Villemur (2015)* that knocking out the *narG2* sequence (Nar2 system; not the upstream non-coding sequence) in strain

JAM1[T] impacted the expression of the *nar1* operon. Contrary to the wild-type strain, this strain JAM1[T] mutant, only expressing the Nar1 system, presented a lag phase (as strain GP59) under the <<AN >> conditions, both for growth and $NO_3^-$ reduction. Furthermore, this mutant cultured under the <<O >> conditions showed the *narG1* transcript levels decreased by 27 times compared to wild type, levels comparable to those of strain GP59 cultured under the same conditions. All these results point out the complexity of the regulation of the *nar1* operon that involves an unknown mechanism linked to the regulation of the *nar2* operon. Part of the puzzle might include the differences in the upstream sequences of *nar2* affecting the affinity of NarL. Other factors that could be involved are FIS and IHF factors, which are well-known DNA binding proteins. There roles comprise compaction, bending of DNA and often bind in regulatory sequences (*Anuchin et al., 2011*). Substitutions observed in the *nar2-nor2* intergenic sequences would affect some of these putative binding sites. It was shown in *E. coli* that NarL and FIS compete on promoters, thus affecting gene expression (*Squire et al., 2009*; *Browning, Butala & Busby, 2019*). However, in our case, how these factors affect the expression of the *nar1* operon 30 kb distant remains unclear.

Even though the control of denitrification happens mostly at the transcriptional level, post transcriptional or translational regulation might play a significant role in the overall process. The involvement of small regulatory bacterial RNA (sRNAs) has been explored in the denitrification pathway. It has been suggested that a sRNA plays a role in modulation the denitrification pathway in *Paraccocus denitrificans* (*Gaimster et al., 2019*), and a potential candidate has been isolated in *Pseudomonas aerugonisa* (*Tata et al., 2017*).

## Influence of culture conditions on the transcriptomes

Transcriptomes were derived from replicate cultures of both strains cultured under the <<O >>, <<ON >> and <<AN >> conditions and sampled during the *High period*, and their relative gene expression profiles were compared. Strains JAM1[T] and GP59 share 2802 coding sequences and 11 riboswitches, having highly, if not 100%, identity in their nucleic acid sequences (Data S3). Among these genes and riboswitches, the lowest changes in relative expression profiles were found between the <<O >> and the <<ON >> conditions in strain JAM1[T] cultures (JO/JON, 2.6%; Fig. S2), and between strain JAM1[T] and strain GP59 cultured under the <<O >> conditions (JO/GO, 2.7%; Fig. S2), whereas the highest changes were found between the strain JAM1[T] cultured under the <<AN >> conditions and strain GP59 cultured under the <<O >> and <<ON >> conditions (JAN/GO, JAN/GON, 30%; Fig. S2).

Principal coordinate analysis (PCoA) was performed with the relative transcript profiles of the common genes for the three culture conditions (Fig. 3). PCoA revealed distinct relative transcript profiles between the <<O >> and <<ON >> cultures and the <<AN >> cultures for both strains (explaining by 50.7% variations; $p = 0.001$). PCoA also revealed distinct profiles between strain GP59 and strain JAM1[T] (second axis, explained by 17.7% variations; $p = 0.003$). These results showed the relative expression profiles were strain specific, and that the presence of oxygen had a deep impact on these profiles in both strains.

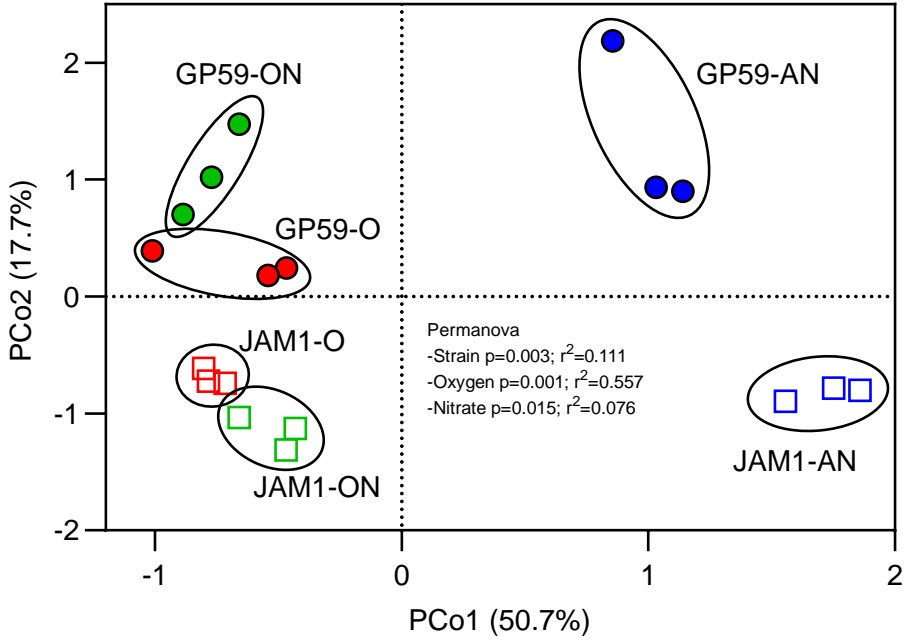

**Figure 3** **Principal coordinate analysis of the relative transcript profiles of the culture replicates.** The relative transcript levels of genes and riboswitches (2813 sequences) common in both genomes were derived from the transcriptomes of culture replicates of strain JAM1[T] and strain GP59 cultured under the <<O >>, <<AN >> and <<ON >> conditions. Principal coordinates analysis was carried out using Bray-Curtis distance calculation, and permutation test (PERMANOVA) were performed with 999 permutations.

In addition to common genes between strain GP59 and strain JAM1[T], strain GP59 has nucleic sequences of its own. It contains two plasmids (96 genes) and a 90.6-kb chromosomic region (119 genes) where *nirK* is located (*Geoffroy et al., 2018*). The culture conditions did not significantly affect the relative transcript levels of all plasmidic genes. However, genes in the 90.6-kb region were 2.5 times ($p < 0.01$) higher in the overall relative transcript levels under the <<O >> and <<ON >> conditions compared to the <<AN >> conditions (Data S3), which may have influenced the expression pattern of *nirK* (see below).

Examining the transcript reads associated with RNA extracted from strain GP59 cultures, we found high proportion of reads not associated with its genome. Transcript reads that did not align with strain GP59 genome and plasmids consisted between 8 to 83% of total reads derived from strain GP59 cultures. *De novo* assembly of these unaligned reads showed three long transcripts that composed 57 to 94% of the unaligned reads, with homology to bacteriophage affiliated to Cystovirus (Table 3, Data S4). These phages are double stranded RNA that were found in bacteria (mainly in *Pseudomonas* species) from diverse environments (*Mäntynen, Sundberg & Poranen, 2018*; *Gottlieb & Alimova, 2023*). No such reads were found in the transcriptomes of strain JAM1[T] cultures. These transcripts probably correspond to the unusual long transcripts apparent on agarose gel electrophoresis of our RNA preparation of strain GP59 (Fig. S1). Presence of these transcripts strongly

**Table 3** Sequence similarity between the metagenomic assembled transcripts and the RNA genome of the Cystovirus *Pseudomonas* phage phi6.

| Gene names | Phage phi-6 | | | MAT | |
| --- | --- | --- | --- | --- | --- |
| | Length (a.a.) | Protein names | | Length (a.a.) | Similarity |
| Segment L (6.4 kb) | | | | (6.9 kb) | |
| P14 | 62 | Protein | | P14 | nsf |
| P7 | 161 | Assembly protein P7 | | 162 | 72% |
| P2 | 665 | RNA-directed RNA polymerase | | 661 | 72% |
| P4 | 332 | Packaging enzyme P4 | | 330 | 62% |
| P1 | 769 | Major inner protein P1 | | 800 | 53% |
| Segment M (4.0 kb) | | | | (3.2 kb) | |
| P8 | 149 | Major outer capsid protein | | | nsf |
| P12 | 195 | Morphogenetic protein | | | nsf |
| P9 | 90 | Major envelope protein | | | nsf |
| P5, P11 | 220 | Peptidoglycan hydrolase gp5 | | 241 | 51% |
| Segment S (2.9 kb) | | | | (3.4 kb) | |
| P10 | 42 | Envelope protein P10 | | | nsf |
| P6 | 168 | Fusion protein P6 | | 179 | 69% |
| P3 | 648 | Spike protein P3 | | 617 | 49% |
| P13 | 72 | Protein P13 | | | nsf |

**Notes.**
Reads that did not align with strain GP59 genome and plasmids were assembled, generating three long metagenomic assembled transcripts (MAT), as indicated by the number of kilobase (kb). Deduced amino acid sequences of open reading frames (ORFs) were examined by BlastP.
Length, number of amino acid (a.a.) deduced from the ORFs; nsf, no similarity found.
Data for the Cystovirus *Pseudomonas* phage phi6 were retrieved from https://viralzone.expasy.org/586.

suggest that strain GP59 releases this phage. Such phage was probably present when we originally isolated strain GP59, as these long transcripts were always present in our RNA extracts (resistant to DNAse treatments), despite inocula were made of a single colony. We never encountered apparent lysis in our cultures, neither seen significant impact on strain GP59 growth. Cystoviruses use the translational machinery of the host for the synthesis of their proteins directly from the viral genome, and possess their own RNA-directed RNA polymerase for genome replication (*Alphonse & Ghose, 2017*). These features suggest that the Cystovirus present in strain GP59 would not have affected the host transcriptional machinery but could have impacted the level of host proteins.

### Influence of the culture conditions on the relative expression of denitrification genes

Figure 4 illustrates fold changes (FC) in the relative transcript levels of *nirK*, *nor1*, *nor2* and *nos*, and other regulatory genes, of all pairwise comparisons (15) between the three conditions respective to each strain (*e.g.*, strain GP59 *versus* <<AN >>, <<ON >>, <<O >> conditions), and between strains respective to the three conditions (*e.g.*, strain JAM1[T] <<AN >> conditions *versus* strain GP59 <<O >> conditions).

Genes expressing the Nor1 system and its regulator NorRE had their relative transcript levels between 2 to 5 times higher under the <<AN >> conditions relative to the <<O

| | Locus tag | GO/GAN | GO/JAN | JO/JAN | JO/GAN | GON/GAN | GON/JAN | JON/JAN | JON/GAN | GO/GON | JO/JON | GO/JON | JO/GON | JAN/GAN | JO/GO | JON/GON |
|---|---|---|---|---|---|---|---|---|---|---|---|---|---|---|---|---|
| *nirK* | CDW43_RS15160 | 19.8 | na | na | na | 7.5 | na | na | na | 2.6 | na | na | na | na | na | na |
| *nor1CBDQ* | Q7A_431-4 | -3.4 | -3.0 | -2.4 | -3.3 | -2.4 | -2.1 | -2.1 | -2.9 | -1.4 | -1.2 | 1.1 | -1.8 | -1.2 | -1.3 | -1.5 |
| *nor1RE* | Q7A_435-6 | -5.1 | -4.1 | -3.9 | -4.7 | -4.9 | -3.8 | -3.0 | -3.6 | -1.0 | -1.3 | -1.3 | -1.0 | -1.2 | 1.0 | 1.2 |
| *nor2CBQD* | Q7A_485-8 | 1.3 | 1.6 | 1.4 | 1.0 | 1.9 | 2.5 | 2.6 | 1.9 | -1.5 | -1.8 | -1.4 | -1.9 | -1.3 | -1.2 | -1.1 |
| *nosR* | Q7A_458 | -1.8 | -3.8 | -3.6 | -1.7 | -1.9 | -3.9 | -3.2 | -1.5 | 1.0 | -1.1 | -1.2 | 1.1 | 2.2 | 1.1 | 1.2 |
| *nosZnosDFYL* | Q7A_459-64 | 1.1 | -1.3 | 1.1 | 1.5 | 1.1 | -1.4 | 1.0 | 1.5 | 1.1 | 1.1 | -1.2 | 1.4 | 1.5 | 1.3 | 1.3 |
| | | | | | | | | | | | | | | | | |
| cAMP Crp | Q7A_386 | -1.7 | 1.8 | 1.5 | -2.0 | -2.2 | 1.4 | -1.1 | -3.2 | 1.3 | 1.6 | 1.9 | 1.1 | -3.0 | -1.2 | -1.5 |
| cAMP Crp | Q7A_945 | 1.0 | 1.6 | 1.5 | 1.2 | 1.3 | 1.7 | 1.0 | 1.3 | 1.3 | 1.0 | 1.0 | 0.9 | 0.8 | 0.9 | 0.9 |
| cAMP Crp | Q7A_1583 | 0.7 | 0.8 | 0.9 | 0.8 | 1.3 | 1.3 | 1.0 | 0.9 | 0.9 | 0.9 | 0.8 | 0.7 | 0.9 | 1.1 | 0.8 |
| cAMP Crp | Q7A_1671 | 0.6 | 1.2 | 1.0 | 0.7 | 1.1 | 1.7 | 1.1 | 0.8 | 0.7 | 0.9 | 1.1 | 0.6 | 0.8 | 0.8 | 0.6 |
| *nnrS1* | Q7A_66 | -3.3 | -6.4 | -8.7 | -4.3 | -2.4 | -4.7 | -5.3 | -2.7 | -1.4 | -1.6 | -1.2 | -1.9 | 2.0 | -1.4 | -1.2 |
| *nsrR1* | Q7A_67 | -3.7 | -4.9 | -6.0 | -4.5 | -2.7 | -3.4 | -4.7 | -3.5 | -1.4 | -1.3 | -1.0 | -1.8 | 1.4 | -1.3 | -1.4 |
| *dnrN/ytfE* | Q7A_68 | -10.1 | -13.8 | -15.9 | -11.6 | -4.9 | -6.6 | -12.5 | -9.1 | -2.1 | -1.3 | -1.1 | -2.4 | 1.4 | -1.2 | -1.9 |
| *nnrS2* | Q7A_438 | -2.1 | -2.0 | -2.3 | -2.3 | -1.9 | -1.8 | -2.0 | -2.1 | -1.1 | -1.1 | 1.0 | -1.3 | -1.0 | -1.2 | -1.1 |
| *nnrS3* | Q7A_1801 | -2.0 | -2.8 | -3.1 | -2.2 | 4.4 | 3.1 | -1.6 | -1.2 | -8.6 | -1.9 | -1.7 | -9.7 | 1.4 | -1.1 | -5.2 |
| *nsrR2* | Q7A_409 | 1.4 | 1.2 | 1.6 | 1.7 | 2.2 | 1.7 | 1.7 | 1.8 | -1.5 | -1.1 | -1.4 | -1.1 | 1.1 | 1.3 | -1.1 |
| *hmp1* | Q7A_410 | -1.7 | 1.3 | -2.2 | -1.6 | -1.6 | 2.0 | -1.3 | -1.8 | -1.0 | -1.7 | 1.3 | -1.7 | -1.8 | -1.2 | -1.9 |
| *hmp2* | Q7A_1974 | -1.4 | -3.2 | -1.1 | -2.3 | 1.0 | 1.1 | 1.0 | -1.3 | -1.5 | 1.1 | -1.8 | -3.7 | 1.4 | -1.0 | -2.1 |
| *iscRSA* | Q7A_1499-1501 | -3.3 | -3.9 | -3.8 | -3.2 | 1.0 | -1.2 | -2.5 | -1.9 | -3.0 | -1.6 | -1.5 | -3.2 | 1.2 | -1.1 | -2.1 |
| *sufA/RBCDSEX* | Q7A_138/1533-39 | -2.0 | -3.9 | -3.4 | -3.2 | -2.3 | -1.9 | -2.2 | -2.0 | -3.2 | -1.6 | -1.6 | -2.0 | 1.2 | -1.0 | -1.4 |

**Figure 4   Fold changes in the relative transcript levels of denitrification genes and regulatory genes.** Genes that were significantly differentially expressed were identified by EdgeR (v 3.36.0). Locus tags are from strain JAM1[T] GenBank annotation (CP003390.3) except *nirK* that is from strain GP59 GenBank annotations (CP021974.1). Values correspond to fold changes (FC) between one condition to the other. In GO/GAN, for example, positive values refer to genes with relative transcript levels higher in the <<O >> conditions than those in the <<AN >> conditions, and negative values refer to the opposite. FC > 2 (red) or < −2 (blue) that have FDR < 0.05 are highlighted by colors. G, strain GP59; J, strain JAM1[T]. O, AN, ON: <<O >>, <<AN >> and <<ON >> conditions, respectively.

>> and <<ON >> conditions in both strains. *nor2* did not show substantial changes in its relative transcript levels in all culture conditions. Despite higher relative transcript levels of *nosR* in strain JAM1[T] cultured under the <<AN >> conditions, no changes were observed with the *nos* gene cluster for both strains in all culture conditions (Fig. 4). *nirK* will be discussed below.

## Influence of the culture conditions on the relative expression of regulatory genes

In addition to *fnr*, four ORFs encoding protein with cAMP-binding domain of CRP or a regulatory subunit of cAMP-dependent protein kinases were found, which may be related to function associated with NnrR or DnrD (*Körner, Sofia & Zumft, 2003*), regulators involved for expression of key denitrification genes (*Honisch & Zumft, 2003*; *Mesa et al., 2003*). No substantial changes (if any) in their relative transcript levels were observed in all conditions between strains for the four cAMP CRP genes (Fig. 4).

Several genes encoding factors involved in NO-response were found in both genomes. This includes (Fig. 4): three *nnrS* genes (*nnrS1* to *nnrS3*), two *hmp* genes (*hmp1* and *hmp2*), two *nsrR* genes (*nsrR1* and *nsrR2*) and the *dnrN/YtfE* gene. Complex and multifaceted response to NO is coordinated by the NO sensitive repressor NsrR (*Volbeda et al., 2017*). The intrinsic reactivity of iron-sulfur (Fe-S) clusters toward NO by NsrR functions as sensor-regulators (*Kennedy, Anthoinie & Beinert, 1997*; *Spiro, 2006*; *Crack et al., 2014*). NnrS is a heme-containing protein, expressed under denitrifying conditions and under the

control of DnrD or NnrR (*Glockner & Zumft, 1996*; *Bartnikas et al., 2002*). NnrS would be one factor that protects cellular iron pool from the formation of dinitrosyl iron complex, and thus avoid NO to inhibit iron-sulfur protein function (*Stern et al., 2013*). In *E. coli*, *ytfE* is under the control of the regulator NsrR and has been shown to protect iron-sulfur cluster-containing proteins (*Justino et al., 2006*; *Justino et al., 2007*; *Overton et al., 2008*). Results from *Crack et al. (2022)* showed that in *E. coli*, YtfE act as NO-forming $NO_2^-$ reductase which allow NO to be detected by NsrR for stimulating other genes such as *hmp* encoding a NO dioxygenase (flavohemoprotein) that oxidizes NO to $NO_3^-$ under oxic conditions (*Poole, 2020*), or the genes clusters encoding Isc and Suf systems involved in production of Fe-S cluster under stress conditions (*Blanc, Gerez & Choudens, 2015*; *Blahut et al., 2020*).

A gene cluster containing *nnrS1*, *nsrR1* and *dnrN/YtfE* had their relative transcript levels 2.4 to 16 times higher in both strains cultured under the <<AN >> conditions than under the <<O >> and <<ON >> conditions. No substantial differences in the relative transcript levels were observed with *nnrS2*, *nsrR2* and the two *hmp* genes between all conditions in both strains. Higher relative transcript levels were observed with the Suf and Isc systems in some cultures for both strains cultured under the <<AN >> and <<ON >> conditions (Fig. 4). All these results suggest that both strains sense the presence of NO in the <<AN >> cultures. Although this was expected in strain GP59 cultures, it remains unclear in strain JAM1[T] cultures because of the lack of NO-forming $NO_2^-$ reductase. It may be related to the anoxic conditions and the presence of $NO_3^-$ in the expectation of NO production.

The most striking results were found with the relative transcript pattern of *nnrS3* between strain GP59 and strain JAM1[T]. This gene had relative transcript levels in strain GP59 cultured under the <<ON >> conditions 9 to 10 times higher than those in strain GP59 and strain JAM1[T] cultured under the <<O >> conditions (Fig. 4: JO/GON, GO/GON), and it was 5 times higher than those in strain JAM1[T] cultured also under the <<ON >> conditions (Fig. 4: JON/GON). Even compared to the <<AN >> conditions, these levels were 4 times higher (Fig. 4: GON/GAN) in favor of the <<ON >> conditions in strain GP59. No difference was observed in strain JAM1[T] between the <<O >> and the <<ON >> conditions (Fig. 4: JO/JON). As NnrS senses NO, our results suggest that NO is generated in strain GP59 cultured under the <<ON >> conditions but because the presence of $O_2$, NO would be transformed back to $NO_3^-$ by the flavohemoproteins Hmp (NO dioxygenase). This reaction could explain why $NO_3^-$ reduction was not observed under the <<ON >> conditions in strain GP59 cultures, even though *nar1* and *nirK* showed significant transcript levels. We therefore hypothesize that the reduction of $NO_3^-$ to NO is concomitant with the oxidation of NO to $NO_3^-$ by Hmp in strain GP59 under oxic conditions.

## Relative expression of *nirK*

The relative transcript levels of *nirK* in strain GP59 cultures were 7.5 and 20 times higher in the <<ON >> and <<O >> conditions, respectively, relative to the <<AN >> conditions, which concurs with the RT-qPCR assays performed with RNA extracted

during the *High period.* This expression pattern was unexpected. Upstream of *nirK* are potential FNR and NarL DNA binding sites (Data S2). Therefore, its expression should be regulated like the other denitrification genes. During the *Low period*, this was the case with higher transcript levels under the <<AN >> conditions compared to the other conditions. During the *High period* and under the <<AN >> conditions, the denitrification pathway is probably expressed enough and does not require further gene stimulation of *nirK*. Under the <<ON >> conditions, as we hypothesized before, $NO_3^-$ and $NO_2^-$ would be reduced in NO then oxidized back to $NO_3^-$, generating a sort of loop, which may explain why gene stimulation of *nirK* is still operating. However, under the <<O >> conditions, the expression pattern of *nirK* is puzzling, because of the absence of $NO_3^-$. What may cause the higher expression of *nirK* during the *High period* under the <<O >> condition remains unclear. Strain GP59 cultures may have encountered nitrosative-stresses under the <<O >> conditions. Presence of $NH_4^+$ in the *Methylophaga* 1403 medium (3.8 mM) is the only source of inorganic N in these conditions. However, genes involved in the transformation of $NH_4^+$ to $NO_2^-$ or $NO_3^-$ are absent in the genome. Another possibility is the abiotic transformation of $NH_4^+$ or organic N compounds generating $NO_3^-$, $NO_2^-$ or NO (*Doane, 2017*). However, the relative expressions of most of the $NO_2^-$/NO reductase regulators (*nsrR, nnrS, dnrN/ytfE*) were higher under the <<AN >> conditions than under the <<O >> conditions. *nirK* is located in a particular chromosomic region, between two prophages, and probably acquired by horizontal transfer as suggested by *Geoffroy et al. (2018)*. Transcriptomes revealed that the relative transcript levels of genes in this region are overall higher under the <<O >> conditions than under the <<AN >> conditions during the *High period* (Data S3), which could have impacted the *nirK* expression.

## CONCLUSIONS

By exploring genes expression, this study shows how two strains of the same species could respond differently and adjust their denitrification pathway according to the environmental conditions. Figure 5 illustrates the proposed mechanisms deduced from our results, in conjunction with the literature, that would explain the denitrification dynamics of each strain. Under the <<O >> conditions that prevailed in the inocula, the Nar1 and Nar2 systems, including the $NO_3^-$ transporters, in strain GP59 cultures are expressed at very low levels. Under the <<AN >> conditions, induction of both systems occurred, which requires some time (latency) before the denitrification pathway is operational. Under the <<ON >> conditions, Nar1 is induced at a low level, similar of what was observed in strain JAM1[T] cultured under the same conditions. These low levels of the Nar systems and NirK are sufficient to generate NO (as indicated by *nnrS3*). Instead to proceed through the NO reductase (Nor systems) that may not be operational because of the oxic conditions, the flavohemoprotein (Hmp) transforms NO back to $NO_3^-$. For strain JAM1[T], the Nar1 system is already expressed in the inocula that allows immediate action on $NO_3^-$ and the accumulation of $NO_2^-$.

The distinct physiological characters of strain JAM1[T] and strain GP59 could be linked to the biofilm environment in the Biodome denitrification system where they were isolated.

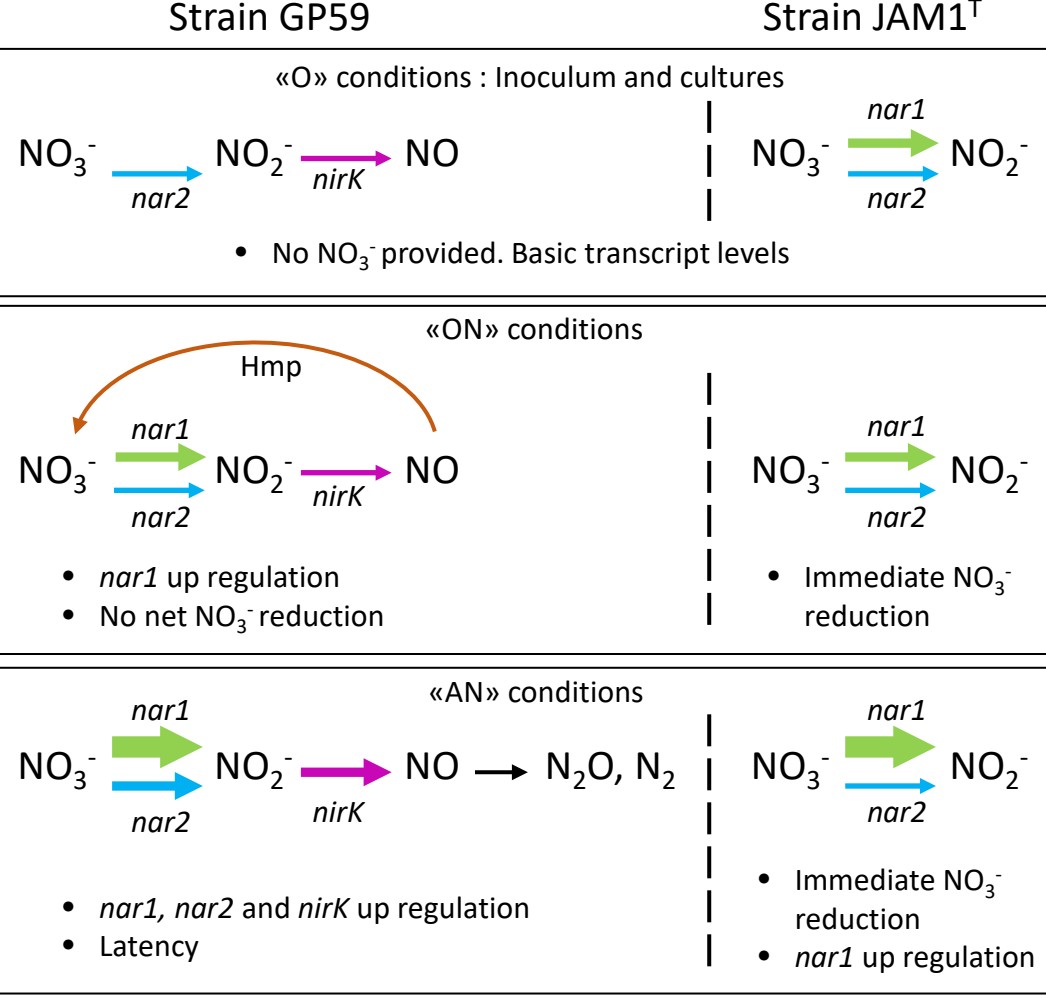

**Figure 5** **Proposed mechanism of denitrification by strain JAM1[T] and strain GP59 during the *Low period*.** Thickness of the arrows represents the gene transcript levels as determined in Fig. 1.

Constitutive expression of the Nar1 system allows strain JAM1[T] to thrive immediately on $NO_3^-$, which is an advantage against other "typical" denitrifiers such as strain GP59 or *Hyphomicrobium nitrativorans* (another major strain isolated from the biofilm), especially as strain JAM1[T] does not reach high level of growth under anoxic conditions.

These different physiological behaviors of strains JAM1[T] and GP59 that belong to the same species are intriguing as both strains originated from the same microbial community of one denitrification reactor. We used to see complex microbial community as an amalgam of different species, but subpopulations of these species could have important impact on the evolution of this community. Globally, the present study and our previous reports on these strains suggest that studying microbial community to identify and isolate the main species in a bioprocess (or any environment) would also have to be extended at the subpopulation species' level. Understanding the mechanisms underlying these differences would provide

indication on the dynamics of these subpopulations in the microbial community upon environmental changes and thus how this community evolved in providing efficient denitrifying activities.

## ACKNOWLEDGEMENTS

We would like to thank Philippe Constant from INRS Centre Armand-Frappier Santé Biotechnologie to carry out PERMANOVA.

### Funding

This research was supported by a grant to Richard Villemur from the Natural Sciences and Engineering Research Council of Canada # RGPIN-2016-06061. The funders had no role in study design, data collection and analysis, decision to publish, or preparation of the manuscript.

### Grant Disclosures

The following grant information was disclosed by the authors:
Natural Sciences and Engineering Research Council of Canada: # RGPIN-2016-06061.

### Competing Interests

The authors declare there are no competing interests.

### Author Contributions

- Livie Lestin conceived and designed the experiments, performed the experiments, analyzed the data, prepared figures and/or tables, authored or reviewed drafts of the article, and approved the final draft.
- Richard Villemur conceived and designed the experiments, analyzed the data, prepared figures and/or tables, authored or reviewed drafts of the article, and approved the final draft.

### DNA Deposition

The following information was supplied regarding the deposition of DNA sequences:

The RNAseq reads from strain JAM1T and strain GP59 cultured under the <<N>> conditions are available at NCBI SRA: PRJNA525230; SRX5461036, SRX5461037, SRX5461044, SRX5461045, SRX5461047, SRX5461048.

For those cultured under the <<O>> and <<ON>> conditions, RNAseq reads are available at: PRJNA1072961; SAMN39755713 to SAMN39755725.

### Data Availability

The data is available in the Supplemental Files.

## Supplemental Information

Supplemental information for this article can be found online at http://dx.doi.org/10.7717/peerj.18361#supplemental-information.

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
