# Peer review of "The bacterial strains JAM1T and GP59 of the species Methylophaga nitratireducenticrescens differ in their expression profiles of denitrification genes in oxic and anoxic cultures"

_PeerJ, doi:10.7717/peerj.18361_

## Round 0.1 · original submission · Minor Revisions

Dear Dr. Lestin and Villemur:

Thanks for submitting your manuscript to PeerJ. I have now received three independent reviews of your work, and as you will see, the reviewers raised some minor concerns about the manuscript. Despite this, these reviewers are optimistic about your work and the potential impact it will have on research studying mechanisms denitrification in Methylophaga nitratireducenticrescens. Thus, I encourage you to revise your manuscript, accordingly, considering all the concerns raised by the reviewers.

There are not too many suggestions; thus, it should not take much effort to address these concerns to greatly improve your manuscript.

I look forward to seeing your revision, and thanks again for submitting your work to PeerJ.

Good luck with your revision,

-joe

·

Basic reporting

The authors Lestin and Villemur used clear and professional English throughout their manuscript and I could not find any mistakes in spelling or grammar.

Remarks here:
The authors use the abbreviations ‘O’, ‘ON’ and ‘N’ for the conditions during the incubations (oxic without NO3-, oxic with NO3- and anoxic with NO3-, respectively). I would suggest to change ‘N’ to ‘AN’ for anoxic with NO3- throughout the manuscript. Using just ‘N’ made me go back and forth in the manuscript to check which abbreviations stands for which conditions. I think changing this to ‘AN’ might make it a bit easier.

Line 198: Can the authors once mention in the text what the abbreviations FIS, IHF and FNR stand for. E.g. FIS - Factor for Inversion Stimulation

Line 470: strains instead of stains



The manuscript includes sufficient introduction and background and the relevant literature is cited.

Only remark here:
Line 53: Methylophaga species can also use volatile organic compounds such as DMS as carbon and energy source for methylotrophic growth.


The manuscript is clearly structured. The Figures are in good quality, appropriately described and labelled.

Remarks here:
Figure 1: Please adjust all axes similar (starting at 0 ending at 4). I thought about another possibility to present this data in an easier way (easier to read and grasp), however since you have two different organisms, two different sampling times (Low versus High) and 3 different incubation conditions, I came to the conclusion that this is probably the best way to present the data in Figure 1. What do the letters a, b, c and so on in the Figure mean? Please add this to the Figure legend. I read that they stand for significantly differences, however it is not clear what the different letters mean. Maybe the authors can provide on example in the legend.

Figure 3: I think this data could be presented in a better way. The authors could consider changing this graph to volcano plots for the two different Methylophaga strains and the different incubation conditions and sampling points. Volcano plots would immediately make it clear what is up and down regulated and even transcripts/genes of interest to this study could be marked in these plots making it even more interesting.

All raw data have been made available.

The manuscript is self-contained with relevant results to hypotheses.

Experimental design

This manuscript represents original primary research within Aims and Scope of the journal.

The authors did well define their research question and fill the knowledge gap of the differences in the regulation pathway of Methylophaga nitratireducentircrescens JAM1 and GP59 and how this impacts the expression of denitrification genes and ultimately growth.
The research in this study has been conducted in a rigorous manner and to a high scientific standard. The used methods are described with sufficient information to be reproducible by other investigators.

Remarks here:
Lines 227-232: The authors identified reads that did not align to GP59 nor JAM1, is that right? What did these reads belong to? Might they just be contamination? In our lab, we observed sometimes contamination from other sequencing experiments from other studies that had been sequences at the same time at the sequencing company. So before using these sequences for further investigation (de novo assembly and estimation of transcript abundance) I would make sure, what these reads belong to, to exclude that these reads might be just contamination.

Lines 382-395: Similar here, could the reads assigned to a bacteriophage (affiliated to Cystovirus) just be a contamination from the sequencing? If the sequencing was performed in-house, then the authors would know if projects on bacteriophages were sequenced around the same time. If sequencing took place at a company or another provider it might be worthwhile to check if and what other sequencing projects were processed around the same time to eliminate the possibility of this being just a contamination from another sequencing project. If this bacteriophage is indeed from GP59, how much could this have influenced the results? This should be discussed in more depth. Especially between the different conditions. Is anything known how this phage behaves under oxic and anoxic conditions?

Validity of the findings

This study investigates a previous research gap and delivers new data, thoughts and results on how two strains of the same species can respond differently to the same environmental conditions concerning their denitrification pathway.

The manuscript's underlying data have been provided and are robust, statistically sound, and well-controlled. The conclusions are clearly stated, directly linked to the original research question, and limited to results supported by the data.

Additional comments

No additional comments.

·

Basic reporting

The original research article entitled “ The bacterial strains JAM1T and GP59 of the species Methylophaga nitratireducenticrescens differ in their expression profiles of denitrification genes in oxic and anoxic cultures” describes the nitrate reduction potential of two different strains of Methylophaga spp. under variable conditions. The research is well-designed, and the experimental works have been performed and presented with sufficient data, diagrams, and explanations. The manuscript is well-written, in professional English.
The literature review should include the objective for experiment design and the ultimate application.
Figures and tables are well-presented, with the necessary raw data and background information in supplementary files.

Experimental design

1. Differential expression analysis by RT-qPCR of certain denitrification genes have been performed for low and high period both. But the transcriptome analysis was performed only for high period samples. Total transcriptome of low period samples could be analyzed to investigate the transcribed genes when NO3- reduction is minimal.
2. The transcriptome analysis of GP59 had revealed the presence of Mu-like phage genome transcripts at elevated level. These findings indicated the possibility of excess transcription and translation of viral RNAs and limiting the transcription of bacterial RNAs. Experiment could be designed by knocking-out the phage genome from the chromosome to understand the effect of cultural condition on the transcriptome level of GP59.
3. Investigation of translational regulation of transcribed genes of denitrification island can be explored to understand the delay of nitrate reduction in GP59 strain under anoxic conditions.
4. Whole transcriptome and proteome analysis of GP59 under latent condition or, lag phase should be investigated to determine the causes for delayed nitrate reduction in GP59.

Validity of the findings

5. Findings with JAM1T strains show promising results, but complete denitrification was not achievable by the strain under any conditions.

Additional comments

The purpose of these experimental findings is not stated clearly. The future application of this research should be focused on the title and conclusion, so the impact of the work will be well-understood. Lastly, as a suggestion, I want to state that if the purpose is to determine a complete denitrification- consortium for bioremediation and solid waste treatment plants, the combined inoculum of JAM1T and GP59 strains of Methylophaga nitratireducenticrescens species should be investigated.

Reviewer 3 ·

Basic reporting

no comment

Experimental design

The experimental conditions only accounted for aerobic and anaerobic environments, without considering the effects of varying oxygen concentrations on the results. Introducing an additional treatment group that explores these variations could yield even more intriguing findings.

Validity of the findings

no comment

Additional comments

General comment
This manuscript explores the different expression profiles of denitrification genes in two denitrifying bacterial strains of Methylophaga nitratireducenticrescens (JAM1T and GP59). The study employs RT-qPCR and genome-centric metatranscriptomics to uncover the expression responses and potential mechanisms. I believe this research is valuable for readers interested in understanding the functional redundancy and evolutionary mechanisms of these two denitrifying strains, which originate from the same complex communities. However, some errors need to be addressed before the manuscript can be accepted. More detailed comments are provided below.

Specific comments
Lines 39-42: Why the nitrate reductase operon (nar1) was expressed at high levels without nitrate? This operon carries immediately nitrate reduction?
Line 43: Please provide the full name of nnrS3 gene. Why its expression indicated presence of NO?
Lines 47-49: This study mainly focused on the differentiated expression of two strains under the oxic and anoxic conditions. Why theses results were related the evolution?
Lines 274-276: The nirK gene exhibited higher level under the oxic condition without nitrate. This functional gene encoded the nitrite reductase?
Line 306: Please provide the possible regulation strategy of the denitrification genes.
Line 314: This section still focused on the gene expression instead of genome sequences.
Lines 376-378: Why did the authors show the gene expression in this region? Please discuss these results or delete them.
Lines 437-443: These results were comparted under the oxic conditions. What about the nnrS gene in two strains under anoxic conditions?
Lines 444-459: Authors further discussed the nirK gene. Please focus these results in one section.
Figure 3: The significance of this figure for the study is minimal, as it does not effectively clarify the regulatory strategies of the denitrification genes.

---

## Round 0.2 · accepted · Accept

Dear Dr. Lestin and Villemur:

Thanks for revising your manuscript based on the concerns raised by the reviewers. I now believe that your manuscript is suitable for publication. Congratulations! I look forward to seeing this work in print, and I anticipate it being an important resource for groups studying mechanisms of denitrification in Methylophaga nitratireducenticrescens.. Thanks again for choosing PeerJ to publish such important work.

Best,

-joe

·

Basic reporting

After reviewing the revised version and the authors' responses to my initial comments, I am satisfied with the revisions they have made. All of my concerns and suggestions have been addressed satisfactorily, and the manuscript has improved as a result.

I recommend the manuscript for acceptance in its current form.

Experimental design

After reviewing the revised version and the authors' responses to my initial comments, I am satisfied with the revisions they have made. All of my concerns and suggestions have been addressed satisfactorily, and the manuscript has improved as a result.

I recommend the manuscript for acceptance in its current form.

Validity of the findings

After reviewing the revised version and the authors' responses to my initial comments, I am satisfied with the revisions they have made. All of my concerns and suggestions have been addressed satisfactorily, and the manuscript has improved as a result.

I recommend the manuscript for acceptance in its current form.

·

Basic reporting

The revised manuscript is well-written, with proper explanations. Also, necessary background information, data and experimental outputs have been incorporated correctly.

Experimental design

The research question, purpose and investigations are performed rationally and presented accordingly.

Validity of the findings

Findings are valid and acceptable.